# P²IR: Universal Deep Node Representation via Partial Permutation Invariant Set Functions

## Abstract

Graph node representation learning is a central problem in social network analysis, aiming to learn the vector representation for each node in a graph. The key problem is how to model the dependence of each node to its neighbor nodes since the neighborhood can uniquely characterize a graph. Most existing approaches rely on defining the specific neighborhood dependence as the computation mechanism of representations, which may exclude important subtle structures within the graph and dependence among neighbors. Instead, we propose a novel graph node embedding method (namely P²IR) via developing a novel notion, namely *partial permutation invariant set function* to learn those subtle structures. Our method can 1) learn an *arbitrary* form of the representation function from the neighborhood, without losing any potential dependence structures, 2) automatically decide the significance of neighbors at different distances, and 3) be applicable to both homogeneous and heterogeneous graph embedding, which may contain multiple types of nodes. Theoretical guarantee for the representation capability of our method has been proved for general homogeneous and heterogeneous graphs. Evaluation results on benchmark data sets show that the proposed P²IR outperforms the state-of-the-art approaches on producing node vectors for classification tasks.

## 1 Introduction

Graph node representation learning (or graph embedding in some literature) is to learn the numerical representation of each node in a graph by vectors in a Euclidean space, where the geometric relationship reflects the structure of the original graph. Nodes that are "close" in the graph are embedded to have similar vector representations (Cai et al., 2017). The learned node vectors benefit a number of graph analysis tasks, such as node classification (Bhagat et al., 2011), link prediction (Liben-Nowell & Kleinberg, 2007), community detection (Fortunato, 2010), and many others (Hamilton et al., 2017b). In order to preserve the node geometric relations in an embedded space, the similarity/proximity/distance of a node to its *neighborhood* is generally taken as input to different graph embedding approaches. For example, matrix-factorization approaches work on pre-defined pairwise similarity measures (e.g., different order of adjacency matrix) (Zhang et al., 2018). Deepwalk (Perozzi et al., 2014), node2vec (Grover & Leskovec, 2016) and other recent approaches (Dong et al., 2017) consider flexible, stochastic measure of node similarity by the node co-occurrence on short random walks over the graph (Goyal & Ferrara, 2017). Neighborhood autoencoder methods compress the information about a node's local neighborhood that is described as a neighborhood vector containing the node's pairwise similarity to all other nodes in the graph (Wang et al., 2016; Cao et al., 2016; Bojchevski & Günnemann, 2018). Neural network based approaches such as graph convolutional networks (GCN) and GraphSAGE apply convolution like functions on its surrounding neighborhood for aggregating neighborhood information (Kipf & Welling, 2016a; Hamilton et al., 2017a; Tu et al., 2018).

However, most existing methods either explicitly or implicitly restrict the dependence form of each node to its neighbors and also the depth of neighbors. Therefore, some important topology structures within the graph and subtle dependence between neighbors may not be captured in the embedded space. For example, the methods in the family of stochastic walk lose the control of influence of further neighbors in a network since they usually utilize a fixed hyper-parameter as a window size, or need to specify the degree of neighborhood proximity; GCN losses the flexibility of neighborhood information since it fixes the two-layer aggregation function for the representation computation.

In this work, we propose a Partial Permutation Invariant Representation method ($P^2$IR) by developing a new notion of *partial permutation invariant set function*, that can

- learn node representation via a *universal* graph embedding function $f$, without pre-defining pairwise similarity, specifying random walk parameters, or choosing aggregation functions among element-wise mean, a max-pooling neural network, or LSTMs;
- capture the arbitrary relationship of each node to its neighbors;
- automatically decide the significance of nodes at different distances;
- be generally applied to any graphs from simple homogeneous graphs to heterogeneous graphs with complicated types of nodes.

Evaluation results on benchmark data sets show that the proposed $P^2$IR outperforms the state-of-the-art approaches on producing node vectors for classification tasks.

## 2 RELATED WORK

The main difference among various graph embedding methods lies in how they define the "closeness" between two nodes (Cai et al., 2017). First-order proximity, second-order proximity or even high-order proximity have been widely studied for capturing the structural relationship between nodes (Tang et al., 2015b; Yang et al., 2017; Zhu et al., 2018). Comprehensive reviews of graph embedding have been done by Cai et al. (2017); Hamilton et al. (2017b); Goyal & Ferrara (2017); Yang et al. (2017). In this section, we discuss the relevant graph embedding approaches in terms of how node closeness to neighboring nodes is measured, for highlighting our contribution on utilizing neighboring nodes in a most general manner.

**Matrix Analysis on Graph Embedding**   As early as 2011, a spectral clustering method (Tang & Liu, 2011) was proposed to take the eigenvalue decomposition of a normalized Laplacian matrix of a graph as an effective approach to obtain the embeddings of nodes. Other similar approaches choose different similar matrix (from Laplacian matrix) to make a trade-off between modeling the "first-order similarity" and "higher-order similarity" (Cao et al., 2015; Ou et al., 2016; Rossi et al., 2018). Node content information can also be easily fused in the pairwise similarity measure, e.g., in TADW (Yang et al., 2015), as well as node label information, which resulting in semi-supervised graph embedding methods, e.g., MMDW (Tu et al., 2016). Recently, an arbitrary-order proximity preserved network embedding method is introduced in (Zhang et al., 2018) based on matrix eigen-decomposition, which is applied to a pre-defined high-order proximity matrix. For heterogeneous networks, Huang et al. (2017) proposed a label involved matrix analysis to learn the classification result of each vertex within a semi-supervised framework.

**Random Walk on a Graph to Node Representation Learning**   Both deepwalk (Perozzi et al., 2014) and node2vec (Grover & Leskovec, 2016) are outstanding graph embedding methods to solve the node representation learning problem. They convert the graph structures into a sequential context format with random walk (Lovász, 1993). Thanks to the invention of Mikolov et al. (2013) for word representation learning of sentences, deepwalk inherited the learning framework for words representation learning in paragraphs to generate the representation of nodes in random walk context. And then node2vec evolved such the idea with additional hyper-parameters tuning for the trade-off between DFS and WFS to control the direction of random walk. Struc2vec (Ribeiro et al., 2017) also utilizes the multilayer graph to construct the graph node representations. Gao & Huang (2018) propose a self-paced network embedding by introducing a dynamic negative sampling method to select difficult negative context nodes in the training process. Planetoid (Yang et al., 2016) proposed a semi-supervised learning framework by guiding random walk with available node label information. The heterogeneity of graph nodes is often handled by a heterogeneous random walk procedure (Dong et al., 2017), or selected relation pairs (Chang et al., 2015). Tang et al. (2015a) considered the predictive text embedding problem on a large-scale heterogeneous text network and the proposed method is also based on pre-defined heterogeneous random walks.

**Neighborhood Encoders to Graph Embedding**   There are also methods focusing on aggregating or encoding the neighbors' information to generate node embeddings. DNGR (Cao et al., 2016) and

SDNE (Wang et al., 2016) introduce the autoencoder to construct the similarity function between the neighborhood vectors and the embedding of the target node. DNGR defines neighborhood vectors based on random walks and SDNE introduces adjacency matrix and Laplacian eigenmaps to the definition of neighborhood vectors. GraphWave (Donnat et al., 2018) learns the representation of each node's neighborhood via leveraging heat wavelet diffusion patterns. Although the idea of autoencoder is a great improvement, these methods are painful when the scale of the graph is up to millions of nodes. Therefore, methods with neighborhood aggregation and convolutional encoders are involved to construct a local aggregation for node embedding, such as GCN (Kipf & Welling, 2016a;b; Schlichtkrull et al., 2018; van den Berg et al., 2017), FastGCN (Chen et al., 2018), column networks (Pham et al., 2017), the GraphSAGE algorithm (Hamilton et al., 2017a), GAT (Velickovic et al., 2017) and a recent DRNE (Tu et al., 2018) method using layer normalized LSTM to approximate the embedding of a target node by the aggregation of its neighbors' embeddings. The main idea of these methods is involving an iterative or recursive aggregation procedure e.g., convolutional kernels or pooling procedures to generate the embedding vectors for all nodes and such aggregation procedures are shared by all nodes in a graph.

The above-mentioned methods work differently on using neighboring nodes for node representation learning. They require on pre-defining pairwise similarity measure between nodes, or specifying random walk parameters, or choosing aggregation functions. In practice, it takes non-trivial effort to tune these parameters or try different measures, especially when graphs are complicated with nodes in multiple types, i.e., heterogeneous graphs. This work hence targets on making neighboring nodes play their roles in a most general manner such that their contributions are learned but not user-defined. The resultant embedding method has the flexibility to work on any types of homogeneous and heterogeneous graph.

Our proposed method P$^2$IR has a natural advantage on avoiding any manual manipulation of random walking strategies or designs for the relationships between different types of nodes. To the invention of set functions by Zaheer et al. (2017), all existing valid mapping strategies from neighborhood to the target nodes can be represented by the set functions which are learnt by P$^2$IR automatically.

## 3 THE PROPOSED P$^2$IR METHOD

In this section, we first formally define the problem, followed by introducing a new notion - partial permutation invariant function set function. This section ends up with the overall P$^2$IR method.

### 3.1 PROBLEM SETUP

We target on designing graph embedding models for the most general graph that may include $K$ different types of nodes ($k$=1 for homogeneous graphs). Formally, a graph $\mathcal{G} = \{\mathcal{V}, \mathcal{E}\}$, where the node set $\mathcal{V} = \cup_{k=1}^{K} \mathcal{V}_k$, i.e., $\mathcal{V}$ is composed of $K$ disjoint types of nodes. One instance of such a graph is the academic publication network, which includes different types of nodes for papers, publication venues, author names, author affiliations, research domains etc. Given such a graph $\mathcal{G}$, our goal is to learn the embedding vector for each node in this graph.

### 3.2 THE PROPOSED UNIVERSAL GRAPH EMBEDDING MODEL

As we know, the position of a node in the embedded space is collaboratively determined by its neighboring nodes. Therefore, the key issue to address for graph embedding is *how to model the dependence of each node to its neighbors*. Most existing approaches need to pre-specify the neighbors (usually via the distance on the graph), and either explicitly or implicitly define a specific form to characterize the dependence between each node and its neighbors.

We propose a universal graph embedding model that neither requires to pre-define neighbors nor to specify the dependence form between each node and its neighbors. The embedding vector $\boldsymbol{x}^v$ of node $v \in \mathcal{V}_k$ can be represented by its neighbors' embedding vectors via a function $f$

$$\boldsymbol{x}^v = f(\boldsymbol{X}_1^v, \boldsymbol{X}_2^v, \cdots, \boldsymbol{X}_K^v), \quad \forall v \in \mathcal{V}_k, \ \forall k \in \{1, 2, \ldots, K\}$$

where $\boldsymbol{X}_k^v$ is a matrix with column vectors corresponding to the embedding of node $v$'s neighbors in type $k$. Note that the neighbors can include step-1 (or immediate) neighbors, step-2 neighbors, and even further neighbors.

**How to characterize an arbitrary dependence to neighbors?** The key observation is that all neighboring nodes reachable from a target node at the same step are not distinguishable from the view of the target node. Therefore, function $f(\cdot)$ should be a *partially permutation invariant function*. That is, if we swap any elements in a $\boldsymbol{X}_k^v$, the function value remains the same. Unfortunately, the set function is not simply learn-able because the permutation property is hard to guarantee directly.

One straightforward idea to represent the partially permutation invariant function is to define it in the following form

$$f(\boldsymbol{X}_1^v, \cdots, \boldsymbol{X}_K^v) := \sum_{\boldsymbol{P}_1 \in \mathbb{P}_{|\mathcal{V}_1|}} \sum_{\boldsymbol{P}_2 \in \mathbb{P}_{|\mathcal{V}_2|}} \cdots \sum_{\boldsymbol{P}_K \in \mathbb{P}_{|\mathcal{V}_K|}} T(\boldsymbol{X}_1^v \boldsymbol{P}_1, \cdots, \boldsymbol{X}_K^v \boldsymbol{P}_K) \qquad (1)$$

where $\mathbb{P}_n$ denotes the set of $n$-dimensional permutation matrices. $\boldsymbol{X}_1^v \boldsymbol{P}_1$ is to permute the columns in $\boldsymbol{X}_1^v$. It is easy to verify that the function defined in equation 1 is partially permutation invariant, but it is intractable because it involves $\prod_{k=1}^N (|\mathcal{V}_k|!)$ "sum" items.

Our solution of learning function $f$ is based on the following important theorem, which gives a neat way to represent any partial permutation invariant function. The proof is provided in the Appendix.

**Theorem 3.1. [Partial permutation invariant function representation theorem]** *Let $f$ be a continuous real-valued function defined on a compact set $\mathbb{X}$ with the following form*

$$f\left(\underbrace{x_{1,1}, x_{1,2}, \cdots, x_{1,N_1}}_{G_1}, \underbrace{x_{2,1}, x_{2,2}, \cdots, x_{2,N_2}}_{G_2}, \cdots, \underbrace{x_{K,1}, x_{K,2}, \cdots, x_{K,N_K}}_{G_K}\right).$$

*If function $f$ is partially permutation invariant, that is, any permutations of the values within the group $G_k$ for any $k$ does not change the function value, then there must exist functions $h(\cdot)$ and $\{g_k(\cdot)\}_{k=1}^K$ to approximate $f$ with arbitrary precision in the following form*

$$h\left(\sum_{n=1}^{N_1} g_1(x_{1,n}), \sum_{n=1}^{N_2} g_2(x_{2,n}), \cdots, \sum_{n=1}^{N_K} g_K(x_{K,n})\right). \qquad (2)$$

The rigorous proof is provided in Appendix. This result essentially suggests a neat but universal way to represent any partially permutation invariant function. For instance, a popular permutation invariant function widely used in deep learning $\max(\cdot)$ can be approximated in an arbitrary precision by

$$\max(x_1, x_2, \cdots, x_N) \approx h\left(\sum_{i=1}^{N} g(x_i)\right)$$

with $g(z) = [\exp(kz)z, \exp(kz)]$, and $h([z, z']) = z/z'$, as long as $k$ is large enough. This is because

$$\max(x_1, x_2, \cdots, x_N) = \lim_{k \to \infty} \left(\sum_{i=1}^{N} \exp(kx_i)\right)^{-1} \sum_{i=1}^{N} \exp(kx_i) \cdot x_i.$$

Therefore, based on Theorem 3.1, we only need to parameterize $h(\cdot)$ and $\{g_k(\cdot)\}_{k=1}^K$ to learn the node embedding function, which makes the partial permutation function finally learn-able and tractable. To make equation 2 approximate an arbitrary partial permutation invariant function, we only need to ensure that functions $h$ and all $g$'s can represent (or approximate) arbitrary real functions. As we know, deep neural network provides an efficient way to approximate an arbitrary continuous real function, because 3-layer neural networks is able to approximate any function (Cybenko, 1989; Hornik, 1991; Haykin, 1994). We next formulate the embedding model when considering different order of neighborhood.

**How to automatically learn the importance of neighbors at different distances?**
(**1-step neighbors**) From Theorem 3.1, *any* mapping function of a node $v \in \mathcal{V}_k$ can be characterized by appropriately defined functions $\psi_1, \psi_2, \cdots, \psi_K$, and $\phi_k$:

$$\boldsymbol{x}^v = \phi_k\left(\sum_{u \in \Omega_{1,1}^v} \psi_1\left(\boldsymbol{x}^u\right), \sum_{u \in \Omega_{1,2}^v} \psi_2\left(\boldsymbol{x}^u\right), \cdots, \sum_{u \in \Omega_{1,K}^v} \psi_K\left(\boldsymbol{x}^u\right)\right) \quad \forall v \in \mathcal{V}_k, \forall k \in \{1, \ldots, K\},$$

where $\Omega_{n,k}^v := \Omega_n^v \cap \mathcal{V}_k$ denotes the step-$n$ neighbors of node $v$ in node type $k$.

(**Multi-step neighbors**) High order proximity has been shown to be beneficial on generating high quality embedding vectors (Yang et al., 2017). Extending the 1-step neighbor model, we can have the more general model where the representation of each node could depend on immediate (1-step) neighbors, 2-step neighbors, 3-step neighbors, and even infinite-step neighbors.

$$\boldsymbol{x}^v = \phi_k \left( \sum_{n=0}^\infty \alpha_n \sum_{u \in \Omega_{n,1}^v} \psi_1(\boldsymbol{x}^u), \ \sum_{n=0}^\infty \alpha_n \sum_{u \in \Omega_{n,2}^v} \psi_2(\boldsymbol{x}^u), \cdots, \sum_{n=0}^\infty \alpha_n \sum_{u \in \Omega_{n,K}^v} \psi_K(\boldsymbol{x}^u) \right) \quad (3)$$

$$\forall v \in \mathcal{V}_k, \forall k \in \{1, 2, \ldots, K\}.$$

where $\alpha_1, \alpha_2, \cdots, \alpha_\infty$ are the weights for neighbors at different steps. Let $\boldsymbol{A} \in \{0,1\}^{|\mathcal{V}| \times |\mathcal{V}|}$ be the adjacent matrix indicating all edges by 1. We involve the polynomial matrix function on the adjacency matrix $\boldsymbol{A}$ to represent the structures of various steps neighborhood with tuning the weight among 1-step to infinite-step neighbors automatically. If we define the polynomial matrix function $B(\cdot)$ on the adjacent matrix $\boldsymbol{A}$ as $B(\boldsymbol{A}) = \sum_{n=0}^\infty \alpha_n \boldsymbol{A}^n$, we can cast equation 3 into its matrix form

$$\boldsymbol{x}^v = \phi_k \left( \bar{\psi}_1(\boldsymbol{X}_1)[B(\boldsymbol{A})]_{\mathcal{V}_1,v}, \ \bar{\psi}_2(\boldsymbol{X}_2)[B(\boldsymbol{A})]_{\mathcal{V}_2,v}, \ \cdots, \ \bar{\psi}_K(\boldsymbol{X}_K)[B(\boldsymbol{A})]_{\mathcal{V}_K,v} \right) \quad (4)$$

$$\forall v \in \mathcal{V}_k, \forall k \in \{1, 2, \ldots, K\},$$

where $\boldsymbol{X}_k$ denotes the representation matrix for nodes in type $k$, $[B(\boldsymbol{A})]_{\mathcal{V}_k,v}$ denotes the sub-matrix of $B(\boldsymbol{A})$ indexed by column $v$ and rows in $\mathcal{V}_k$, and function $\bar{\psi}$ (with $\bar{\ }$ on the top of function $\psi(\cdot)$) is defined as the function extension

$$\bar{\psi}(\boldsymbol{X}) := [\psi(\boldsymbol{x}_1), \psi(\boldsymbol{x}_2), \cdots, \psi(\boldsymbol{x}_N)],$$

where $\bar{\psi}(\cdot)$ takes the output column vectors from a function $\psi(\cdot)$ with each column of $\boldsymbol{X}$ to form a new matrix with $N$ columns. For a specific node $v$, we apply the scale $|\mathcal{V}_{i,v}|$ as $N$ for the type $i$ neighbors of node $v$ in the equation above. Therefore, with the polynomial matrix function $B(\boldsymbol{A})$, and multiple embedding vector mapping $\bar{\psi}(\cdot)$s, we integrate the subtle structural knowledge of various types and steps neighbors surrounding a node to compute the node representation. Note that the embedding vectors for different types of nodes may be with different dimensions. Homogeneous graph is a special case of the heterogeneous graph with $K = 1$. The above proposed model is thus naturally usable on a homogeneous graph.

To avoid optimizing infinite number of coefficients, we propose to use a 1-dimensional NN function $\rho(\cdot): \mathbb{R} \to \mathbb{R}$ to equivalently represent the function $B(\cdot)$ to reduce the number of parameters based on the following observations

$$B(\boldsymbol{A}) = \boldsymbol{U} \mathrm{diag} \left( \sum_{n=0}^\infty \alpha_n \sigma_1^n, \cdots, \sum_{n=0}^\infty \alpha_n \sigma_N^n \right) \boldsymbol{U}^\top = \boldsymbol{U} \mathrm{diag} \left( \rho(\sigma_1), \cdots, \rho(\sigma_N) \right) \boldsymbol{U}^\top,$$

where $\boldsymbol{A} = \boldsymbol{U} \mathrm{diag}(\sigma_1, \cdots, \sigma_N) \boldsymbol{U}^\top$ is the singular value decomposition (SVD). We parameterize $\rho: \mathbb{R} \mapsto \mathbb{R}$ using 1-dimensional NN function, which allows us easily controlling the number of variables to optimize in $\rho$ by choosing the number of layers and the number of nodes in each layer.

**Remark 3.2.** *Acute readers may worry about the computational complexity of SVD when the size of A is large. In general, the full SVD takes the cubic computational complexity $O(N^3)$ where $N$ is the total number of nodes in the network, which is pretty expensive if $N$ is large. However, since our adjacency matrix of the network is typically a very sparse matrix (the number of edges is typically far less than $O(N^2)$), the complexity of the matrix decomposition can be reduced from $O(N^3)$ to $O(N \times \#e)$ where $\#e$ is the number of edges in the network. Furthermore, we can use the low rank approximation approach (that is, top $k$ eigenvalues and eigenvectors) to approximate the original large sparse matrix. Therefore, the approximated version of the matrix decomposition can be completed in $O(k \times \#e)$ complexity. We apply the low rank approximation in our experiments. In particular, such approximation has been used in large datasets including Pubmed, DBLP, and BlogCatalog. We also tested it on the small datasets including Cora, Citeseer, and Wiki by only using top eigenvalues in the range of top 15% to 50% and did not found obvious drop on the performance.*

### 3.3 THE OVERALL P$^2$IR MODEL

For short, we denote the representation function for $\boldsymbol{x}^v$ in equation 4 by

$$\boldsymbol{x}^v = \mathcal{R}_{\rho,\phi^k,\{\psi^k\}_{k=1}^K}(v) \quad \forall v \in \mathcal{V}_k \,, \forall k \in \{1, 2, \dots, K\}.$$

To fulfill the requirement of a specific learning task, we propose the complete P$^2$IR model by involving a supervised component

$$\min_{\substack{\{\boldsymbol{x}^v\}_{v \in \mathcal{V}}, \rho, \\ \{\phi^k\}_{k=1}^K, \{\psi^k\}_{k=1}^K}} \quad \sum_{k=1}^K \frac{1}{\lambda_k |\mathcal{V}_k|} \sum_{u \in \mathcal{V}_k} \left\| \boldsymbol{x}^u - \mathcal{R}_{\rho,\phi^k,\{\psi^k\}_{k=1}^K}(u) \right\|^2 + \frac{1}{|\mathcal{V}_{\text{label}}|} \sum_{v \in \mathcal{V}_{\text{label}}} \ell(\boldsymbol{x}^v, y^v) \quad (5)$$

where $\mathcal{V}_{\text{label}} \subset \mathcal{V}$ denotes the set of labeled nodes, and $\lambda_k > 0$ balances the representation error and prediction error. The first unsupervised learning component restricts the representation error between the target node and its neighbors with $L_2$ norm since it is allowed to have noise in a practical graph. And the supervised component is flexible to be replaced with any designed learning task on the nodes in a graph. For example, to a regression problem, a least square loss can be chosen to replace $\ell(\boldsymbol{x}, y)$ and a cross entropy loss can be used to formulate a classification learning problem. To solve the problem in equation 5, we apply a stochastic gradient descent algorithm (SGD) to compute the effective solutions for the learning variables simultaneously.

## 4 EXPERIMENTS

This section reports experimental results to validate the proposed method, comparing to state-of-the-art algorithms on benchmark datasets including both homogeneous and heterogeneous graphs, for evaluating the applicability of our proposed method on embedding problem of general graphs.

### 4.1 COMPARISON ON HOMOGENEOUS GRAPHS

We consider the multi-class classification problem over the homogeneous graphs. Given a graph with partially labeled nodes, the goal is to learn the representation for each node for predicting the class for unlabeled nodes.

**Datasets**    We evaluate the performance of P$^2$IR and methods for comparison on five datasets.

- **Cora** (McCallum et al., 2000) is a paper citation network. Each node is a paper. There are 2708 papers and 5429 citation links in total. Each paper is associated with one of 7 classes.
- **CiteSeer** (Giles et al., 1998) is another paper citation network. CiteSeer contains 3,312 papers and 4,732 citations in total. All these papers have been classified into 6 classes.
- **Pubmed** (Sen et al., 2008) is a larger and more complex citation networks compared to previous two datasets. There are 19,717 vertexes and 88,651 citation links. Papers are classified into 3 classes.
- **Wikipedia** (Sen et al., 2008) contains 2,405 online web-pages. The 17,981 links are undirected between pairs of them. All these pages are from 17 categories.
- **Email-eu** (Leskovec et al., 2007) is an Email communication network which illustrates the email communication relationships between researchers in a large European research institution. There are 1,005 researchers and 25,571 links between them. Department affiliations (42 in total) of the researchers are considered as labels to predict.

**Baseline methods**    The compared baseline algorithms are listed below:

- **Deepwalk** (Perozzi et al., 2014) is an unsupervised graph embedding method which relies on the random walk and word2vec method. For each vertex, we take 80 random walks with length 40, and set window size as 10. Since deepwalk is **unsupervised**, we apply a logistic regression on the generated embeddings for node classification.
- **Node2vec** (Grover & Leskovec, 2016) is an improved graph embedding method based on deepwalk. We set the window size as 10, the walk length as 80 and the number of walks for each node is set to 100. Similarly, the node2vec is **unsupervised** as well. We apply the same evaluation procedure on the embeddings of node2vec as what we did for deepwalk.

- **Struc2vec** (Ribeiro et al., 2017) chooses the window size as 10, the walking length as 80, the number of walks from each node as 10, and 5 iterations in total for SGD.
- **GraphWave** (Donnat et al., 2018) chooses the heat coefficient as 1000, the number of characteristic functions as 50, the number of Chebyshev approximations as 100, and the number of steps as 20.
- **WYS (Watch-Your-Step)** (Abu-El-Haija et al., 2017) chooses the learning rate as 0.2, the highest power of normalized adjacency matrix as 5, the regularization co-efficient as 0.1, and uses the "Log Graph Likelihood" as objective function.
- **MMDW** (Tu et al., 2016) is a **semi-supervised** learning framework of graph embedding which combines matrix decomposition and SVM classification. We tune the method multiple times and take 0.01 as the hyper-parameter $\eta$ in the method which is recommended by the authors.
- **Planetoid** (Yang et al., 2016) is a **semi-supervised** learning framework. We set the batch size as 200, learning rate as 0.01, the batch size for label context loss as 200, and mute the node attributes as input while using softmax for the model output.
- **GCN (Graph Convolutional Networks)** (Kipf & Welling, 2016a) chooses the convolutional neural networks into the **semi-supervised** embedding learning of graph. We eliminate the node attributes for fairness as well.
- **GAT (Graph Attention Networks)** (Velickovic et al., 2017) chooses the learning rate as 0.005, the co-efficient of the regularization as 0.0005, and the number of hidden units as 64. To make the comparison fair, we mute the node attributes in the training of GAT as well.

**Experiment setup and results** For fair comparison, the dimension of representation vectors is chosen to be the same for all algorithms (the dimension is $64$). The hyper-parameters are fine-tuned for all of them. The details of P$^2$IR for multi-class case are as follows.

- **Supervised Component:** *Softmax* function is chosen to formulate our supervised component in equation 5 which is defined as $\sigma : \mathbb{R}^K \to \{\boldsymbol{z} \in \mathbb{R}^K | \sum_{i=1}^{K} z_i = 1, z_i > 0\}$. For an arbitrary embedding $\boldsymbol{x} \in \mathbb{R}^d$, we have the probability term as $\mathrm{P}(y = j|\boldsymbol{x}) = \frac{\exp(\boldsymbol{w}_j^\top \boldsymbol{x} + b_j)}{\sum_i^K \exp(\boldsymbol{w}_i^\top \boldsymbol{x} + b_i)}$ for such node to be predicted as class $j$, where $(\boldsymbol{w}_j \in \mathbb{R}^d, b_j \in \mathbb{R})$ is a classifier for class $j$. Therefore, the whole supervised component in equation 5 is $-\sum_{v \in \mathcal{V}_{\text{label}}} \sum_j^K y_j \log \mathrm{P}(y = j|\boldsymbol{x}_v) + \lambda_w \mathcal{R}(\boldsymbol{w})$, where $\mathcal{R}(\boldsymbol{w})$ is an $L_2$ regularization for $\boldsymbol{w}$ and $\lambda_w$ is chosen to be $10^{-3}$.
- **Unsupervised embedding mapping Component:** We design a two-layer NN with hidden dimension 64 to form the mapping from embeddings of neighbors to the target node and we also form a two-layer 1-to-1 NN with a 3 dimensional hidden layer to construct the matrix function for the adjacency matrix. We pre-process the matrix $\boldsymbol{A}$ with an eigenvalue decomposition where the procedure preserve the highest 1000 eigenvalues in default. The balance hyper-parameter $\lambda_1$ is set to be $10^{-3}$.

Table 1: Accuracy (%) of multi-class classification experiments in homogeneous graphs

| | training% | 10.00% | 20.00% | 30.00% | 40.00% | 50.00% | 60.00% | 70.00% | 80.00% | 90.00% |
|---|---|---|---|---|---|---|---|---|---|---|
| Cora | deepwalk | 72.73 ±2.02 | 75.50 ±1.12 | 76.72 ±0.99 | 77.34 ±0.52 | 78.38 ±0.82 | 78.41 ±1.47 | 78.82 ±1.12 | 79.15 ±1.04 | 79.70 ±2.32 |
| | node2vec | 73.46 ±0.87 | 75.70 ±1.33 | 76.45 ±1.11 | 78.07 ±1.03 | 78.42 ±1.14 | 78.65 ±1.46 | 78.94 ±1.43 | 79.08 ±1.44 | 81.04 ±0.99 |
| | GraphWave | 30.27 ±0.11 | 30.44 ±0.29 | 30.41 ±0.30 | 30.11 ±1.02 | 30.47 ±1.93 | 30.29 ±0.59 | 30.47 ±0.45 | 31.09 ±0.73 | 31.19 ±3.15 |
| | struc2vec | 26.89 ±0.71 | 28.47 ±0.32 | 29.69 ±0.39 | 29.53 ±1.02 | 30.06 ±2.01 | 30.01 ±0.88 | 30.34 ±0.59 | 31.09 ±0.66 | 31.19 ±3.09 |
| | WYS | 70.24 ±1.55 | 72.82 ±0.99 | 74.17 ±1.00 | 74.79 ±0.30 | 75.51 ±1.37 | 75.18 ±0.92 | 75.96 ±1.29 | 76.01 ±1.04 | 77.11 ±2.53 |
| | MMDW | 74.88 *±0.23 | 79.18 *±0.10 | 81.20 *±0.11 | 82.19 *±0.25 | 83.10 *±0.41 | 84.62 *±0.09 | 85.54 *±0.44 | 85.27 *±0.22 | 87.82 *±0.37 |
| | planetoid | 40.87 ±15.61 | 52.63 ±0.44 | 52.23 ±1.47 | 52.07 ±0.55 | 53.06 ±1.99 | 51.86 ±1.90 | 51.91 ±1.09 | 51.29 ±2.76 | 54.32 ±3.48 |
| | GCN | 59.10 ±1.48 | 67.08 ±1.98 | 73.06 ±1.07 | 75.61 ±1.02 | 79.08 ±1.23 | 80.11 ±1.15 | 82.61 ±1.00 | 84.29 ±1.88 | 85.18 ±1.66 |
| | GAT | 29.57 ±2.11 | 30.49 ±0.67 | 29.91 ±0.58 | 29.85 ±0.81 | 30.44 ±0.78 | 30.22 ±1.73 | 30.53 ±0.79 | 30.74 ±2.33 | 30.55 ±1.41 |
| | **P$^2$IR** | **76.28** **±1.35** | **80.10** **±1.36** | **82.95** **±1.44** | **83.99** **±1.14** | **85.13** **±0.81** | **85.95** **±0.71** | **86.87** **±0.97** | **87.84** **±0.35** | **88.59** **±1.58** |
| Citeseer | deepwalk | 46.52 ±1.45 | 49.18 ±0.49 | 49.28 ±0.72 | 50.41 ±0.87 | 50.43 ±1.27 | 50.79 ±0.99 | 49.63 ±0.75 | 50.06 ±1.89 | 52.33 ±0.89 |
| | node2vec | 46.38 ±1.32 | 47.65 ±0.88 | 48.68 ±0.71 | 48.12 ±0.64 | 48.37 ±1.14 | 48.13 ±0.98 | 47.67 ±1.38 | 48.55 ±1.11 | 48.82 ±2.46 |
| | GraphWave | 18.03 ±0.30 | 18.03 ±0.28 | 17.96 ±0.17 | 18.40 ±0.46 | 18.14 ±0.70 | 17.67 ±1.06 | 17.76 ±0.35 | 18.13 ±1.91 | 19.58 ±1.92 |

| Dataset | Method | 10% | 20% | 30% | 40% | 50% | 60% | 70% | 80% | 90% |
|---|---|---|---|---|---|---|---|---|---|---|
| | struc2vec | 17.72 ±0.35 | 17.73 ±0.34 | 17.79 ±0.44 | 17.91 ±0.72 | 18.16 ±0.88 | 17.52 ±1.07 | 17.76 ±0.41 | 18.10 ±2.10 | 18.79 ±1.85 |
| | WYS | 44.23 ±1.56 | 44.14 ±0.42 | 44.77 ±1.15 | 44.38 ±1.13 | 43.84 ±1.31 | 44.15 ±1.26 | 43.63 ±1.53 | 44.32 ±1.75 | 46.59 ±1.44 |
| | MMDW | **55.36** ±**0.60** | **60.98** ±**0.56** | **62.00**\* ±0.17 | **63.89**\* ±0.12 | **66.59**\* ±0.27 | **69.00** ±0.15 | **69.72**\* ±0.82 | **70.40**\* ±0.93 | 70.64 ±0.31 |
| | planetoid | 38.44 ±0.89 | 41.12 ±0.98 | 40.67 ±1.42 | 38.59 ±1.50 | 39.32 ±2.13 | 37.96 ±1.60 | 39.07 ±0.72 | 36.98 ±1.59 | 36.75 ±3.32 |
| | GCN | 39.78 ±0.34 | 47.89 ±1.21 | 53.85 ±1.38 | 57.92 ±1.21 | 62.00 ±0.72 | 64.74 ±1.09 | 67.59 ±1.32 | 68.04 ±1.56 | 73.65\* ±1.03 |
| | GAT | 20.45 ±0.60 | 21.82 ±3.02 | 23.21 ±4.34 | 22.50 ±3.48 | 25.26 ±5.25 | 23.76 ±5.38 | 25.19 ±8.41 | 24.32 ±6.32 | 25.83 ±7.68 |
| | **P²IR** | 53.98\* ±1.64 | 60.34\* ±0.81 | **63.55** ±**0.97** | **66.34** ±**0.63** | **69.41** ±**1.10** | **71.63** ±**0.92** | **73.11** ±**1.82** | **72.99** ±**1.47** | **77.16** ±**1.82** |
| Pubmed | deepwalk | 75.51 ±0.34 | 75.84 ±0.20 | 76.00 ±0.19 | 76.15 ±0.22 | 76.13 ±0.38 | 76.22 ±0.26 | 76.05 ±0.37 | 76.70 ±0.83 | 76.03 ±1.08 |
| | node2vec | 76.17\* ±0.26 | 76.46\* ±0.26 | 76.41\* ±0.17 | 76.67\* ±0.31 | 76.96\* ±0.48 | 76.82\* ±0.42 | 76.69\* ±0.54 | 77.41\* ±0.96 | 76.85\* ±1.29 |
| | GraphWave | 38.80 ±0.66 | 39.22 ±0.90 | 38.87 ±0.31 | 39.17 ±0.35 | 39.31 ±0.44 | 39.17 ±0.43 | 39.21 ±0.39 | 39.35 ±0.46 | 38.77 ±0.79 |
| | struc2vec | - | - | - | - | - | - | - | - | - |
| | WYS | 75.09 ±0.47 | 75.48 ±0.29 | 75.52 ±0.32 | 75.82 ±0.30 | 75.88 ±0.60 | 75.81 ±0.55 | 75.72 ±0.50 | 76.28 ±1.01 | 75.42 ±1.14 |
| | MMDW | 68.57 ±1.01 | 66.18 ±4.13 | 67.39 ±5.02 | 68.95 ±1.53 | 71.84 ±2.93 | 71.44 ±2.33 | 53.65 ±14.69 | 69.11 ±6.45 | 50.90 ±16.62 |
| | planetoid | 40.27 ±0.50 | 40.48 ±0.23 | 40.30 ±0.50 | 40.33 ±0.23 | 40.33 ±0.49 | 40.79 ±0.55 | 40.66 ±0.48 | 40.43 ±0.31 | 40.92 ±1.35 |
| | GCN | 57.17 ±0.44 | 60.92 ±0.70 | 63.70 ±0.33 | 65.76 ±0.35 | 66.96 ±0.43 | 67.80 ±0.65 | 68.06 ±0.61 | 69.39 ±0.30 | 70.24 ±0.95 |
| | GAT | 41.62 ±4.26 | 43.44 ±4.72 | 44.47 ±6.04 | 41.99 ±4.75 | 46.64 ±8.10 | 47.05 ±6.85 | 45.45 ±6.86 | 48.68 ±6.78 | 40.97 ±2.55 |
| | **P²IR** | **76.91** ±**0.73** | **78.50** ±**0.40** | **79.38** ±**0.23** | **80.26** ±**0.27** | **80.54** ±**0.52** | **81.28** ±**0.46** | **81.58** ±**0.37** | **81.99** ±**0.71** | **82.17** ±**0.60** |
| Wikipedia | deepwalk | 45.82 ±1.05 | 51.22 ±1.05 | 52.18 ±0.89 | 54.05 ±1.13 | 54.98 ±0.79 | 54.45 ±1.46 | 56.37 ±2.19 | 56.34 ±2.11 | 57.00 ±2.91 |
| | node2vec | 45.04 ±0.74 | 49.70 ±0.82 | 52.17 ±0.98 | 52.45 ±1.80 | 53.76 ±0.85 | 53.41 ±1.26 | 54.87 ±1.98 | 55.93 ±1.64 | 53.83 ±4.16 |
| | GraphWave | 4.73 ±0.38 | 5.04 ±0.53 | 5.03 ±0.43 | 5.49 ±0.32 | 5.34 ±0.69 | 5.11 ±0.37 | 5.58 ±0.49 | 5.16 ±0.90 | 6.00 ±1.83 |
| | struc2vec | 11.42 ±1.39 | 12.64 ±1.08 | 11.78 ±0.75 | 12.31 ±1.44 | 12.61 ±0.62 | 12.45 ±0.81 | 12.43 ±1.68 | 12.39 ±1.64 | 12.50 ±2.24 |
| | WYS | 45.24 ±1.70 | 48.25 ±1.04 | 49.64 ±0.92 | 49.49 ±1.55 | 50.08 ±1.03 | 49.75 ±1.15 | 50.29 ±2.54 | 51.52 ±1.20 | 50.17 ±2.69 |
| | MMDW | 53.05\* ±0.54 | 59.45\* ±0.35 | 62.85\* ±0.55 | 62.42 ±0.07 | 64.26 ±1.09 | 66.46 ±0.85 | 67.50 ±0.48 | 67.37 ±0.56 | 70.20 ±1.22 |
| | planetoid | 49.78 ±1.81 | 51.44 ±0.93 | 51.61 ±0.55 | 50.35 ±1.38 | 50.32 ±0.59 | 49.89 ±2.02 | 51.00 ±1.89 | 50.50 ±3.01 | 52.03 ±2.60 |
| | GCN | 52.93 ±1.91 | 59.12 ±1.16 | 62.63 ±1.27 | 66.21\* ±0.59 | 67.46\* ±1.07 | 69.00\* ±1.40 | 69.74\* ±2.28 | 71.85\* ±0.67 | 72.58\* ±3.79 |
| | GAT | 16.83 ±0.98 | 15.70 ±2.50 | 16.30 ±1.01 | 16.84 ±0.90 | 15.84 ±1.99 | 15.47 ±3.60 | 15.90 ±1.84 | 17.18 ±2.26 | 15.52 ±2.73 |
| | **P²IR** | **56.02** ±**1.00** | **63.96** ±**0.62** | **65.92** ±**0.59** | **68.99** ±**0.80** | **69.80** ±**0.80** | **71.27** ±**0.89** | **71.57** ±**2.59** | **73.81** ±**1.81** | **74.33** ±**2.85** |
| Email-eu | deepwalk | 28.87 ±2.40 | 42.59 ±2.95 | 48.02 ±2.10 | 52.07 ±0.71 | 56.81 ±1.10 | 57.26 ±2.44 | 60.00 ±2.55 | 59.80 ±5.07 | 58.80 |
| | node2vec | 29.82 ±1.79 | 43.93 ±3.72 | 48.42 ±2.03 | 52.90 ±0.78 | 56.81 ±2.08 | 56.87 ±1.87 | 59.38 ±3.17 | 59.50 ±2.32 | 60.00 ±6.44 |
| | GraphWave | 6.50 ±0.21 | 6.64 ±0.47 | 6.69 ±0.52 | 6.14 ±0.59 | 6.57 ±0.45 | 6.37 ±0.72 | 6.71 ±2.12 | 7.26 ±2.27 | 7.80 ±1.48 |
| | struc2vec | 6.79 ±0.33 | 6.94 ±0.52 | 6.83 ±0.41 | 6.24 ±0.46 | 6.85 ±0.36 | 6.37 ±0.62 | 6.71 ±2.12 | 7.66 ±2.24 | 7.80 ±1.30 |
| | WYS | 46.84 ±1.79 | 53.11 ±2.30 | 54.68 ±3.12 | 56.25 ±1.02 | 61.24 ±3.32 | 60.10 ±2.13 | 59.73 ±3.49 | 60.50 ±1.52 | 60.60 ±5.37 |
| | MMDW | 36.76 ±0.90 | 40.72 ±2.54 | 43.22 ±0.61 | 43.01 ±1.52 | 46.11 ±1.23 | 44.94 ±0.38 | 48.08 ±1.21 | 53.62 ±0.83 | 65.50 ±1.34 |
| | planetoid | 53.50 ±2.60 | 61.26 ±1.13 | 62.75 ±2.39 | 65.19 ±1.09 | 67.02 ±1.72 | 66.41 ±2.46 | 67.51 ±2.14 | 68.44 ±2.90 | 69.06 ±3.95 |
| | GCN | 58.94\* ±3.57 | 64.93\* ±2.33 | 69.13\* ±1.38 | 70.65\* ±2.13 | 72.87\* ±1.55 | 74.13\* ±2.96 | 75.35\* ±1.87 | 76.12\* ±1.99 | 77.60\* ±3.51 |
| | GAT | 10.71 ±0.21 | 9.47 ±2.39 | 10.04 ±1.89 | 9.40 ±2.37 | 10.26 ±0.73 | 10.37 ±1.22 | 11.46 ±2.13 | 12.18 ±2.15 | 14.26 ±6.63 |
| | **P²IR** | **63.05** ±**5.74** | **67.19** ±**3.39** | **71.81** ±**2.39** | **73.23** ±**2.73** | **76.33** ±**1.56** | **75.57** ±**0.96** | **76.41** ±**3.46** | **77.81** ±**2.35** | **79.00** ±**4.53** |

We take experiments on each data set and compare the performance among all baseline methods mentioned above. Since it is the multi-class classification scenario, we use *Accuracy* as the evaluation criterion. The percentage of labeled samples is chosen from $10\%$ to $90\%$ and the remaining samples

are used for evaluation. All experiments are repeated for five times and we report the mean and standard deviation of their performance in the Table 1 . We highlight the best performance for each dataset with **bold** font style and the second best results with a "*". We can observe that in most cases, our method outperforms other methods.

## 4.2 COMPARISON ON HETEROGENEOUS GRAPHS

We next conduct evaluation on heterogeneous graphs, where learned node embedding vectors are used for multi-label classification.

**Datasets** The used datasets include

- **BlogCatalog** (Wang et al., 2010) is a social media network with 55,814 users and according to the interests of users, they are classified into multiple overlapped groups. We take the five largest groups to evaluate the performance of methods. Users and tags are two types of nodes. The 5,413 tags are generated by users with their blogs as keywords. Therefore, tags are shared with different users and also have connections since some tags are generated from the same blogs. The number of edges between users, between tags and between users and tags are about 1.4M, 619K and 343K respectively.

- **DBLP** (Ji et al., 2010) is an academic community network. Here we obtain a subset of the large network with two types of nodes, authors and key words from authors' publications. The generated subgraph includes $27K$ (authors) + $3.7K$ (key words) vertexes. The link between a pair of author indicates the coauthor relationships, and the link between an author and a word means the word belongs to at least one publication of this author. There are 66,832 edges between pairs of authors and 338,210 edges between authors and words. Each node can have multiple labels out of four in total.

**Baseline Methods** To illustrate the validation of the performance of $P^2IR$ on heterogeneous graphs, we conduct the experiments on two stages: (1) comparing $P^2IR$ with Deepwalk (Perozzi et al., 2014) and node2vec (Grover & Leskovec, 2016) on the graphs by treating all nodes as the same type ($P^2IR$ with $K = 1$ in a homogeneous setting); (2) comparing $P^2IR$ with the state-of-the-art heterogeneous graph embedding method, *metapath2vec* (Dong et al., 2017), in a heterogeneous setting. The hyper-parameters of the method are fine-tuned and *metapath2vec++* is chosen as the option for the comparison.

**Experiment Setup and Results** For fair comparison, the dimension of representation vectors is chosen to be the same for all algorithms (the dimension is 64). We fine-tune the hyper-parameter for all of them. The details of $P^2IR$ for multi-label case are as follows.

- **Supervised Component:** Since it is a multi-label classification problem, each label can be treated as a binary classification problem. Therefore, we apply logistic regression for each label and for an arbitrary instance $x$ and the $i$-th label $y_i$, the supervised component is formulated as $l(\boldsymbol{x}, y_i) = \log(1 + \exp(\boldsymbol{w}_i^\top \boldsymbol{x} + b_i)) - y_i(\boldsymbol{w}_i^\top \boldsymbol{x} + b_i)$, where $(\boldsymbol{w}_i \in \mathbb{R}^d, b_i \in \mathbb{R})$ is the classifier for the $i$-th label. Therefore, the supervised component in equation 5 is defined as $\sum_{v \in \mathcal{V}_{\text{label}}} \sum_i l(\boldsymbol{x}, y_i) + \lambda_w \mathbb{R}(\boldsymbol{w}_i)$ and $\mathcal{R}(\boldsymbol{w}_i)$ is the regularization component for $\boldsymbol{w}_i$, where $\lambda_w$ is chosen to be $10^{-4}$.

- **Unsupervised Embedding Mapping Component:** We design a two-layes NN with a 64-dimensional hidden layer for each type of nodes with the types of nodes in its neighborhood to formulate the mapping from embedding of neighbors to the embedding of the target node. We also form a two-layer 1-to-1 NN wth a 3 dimensional hidden layer to construct the matrix function for the adjacency matrix $A$ for the whole graph. We pre-process the matrix $\boldsymbol{A}$ with an eigenvalue decomposition by preserving the highest 1000 eigenvalues of magnitude in default. We denote the nodes to be classified as type 1 and the other type as type 2. The balance hyper-parameter $[\lambda_1, \lambda_2]$ is set to be [0.2, 200].

For the datasets DBLP and BlogCatalog, we carry out the experiments on each of them and compare the performance among all methods mentioned above. Since it is a multi-label classification task, we take *f1-score(macro, micro)* as the evaluation score for the comparison. The percentage of labeled samples is chosen from 10% to 90%, while the remaining samples are used for evaluation. We repeat all experiments for three times and report the mean and standard deviation of their performance in the Table 2. We can observe that in most cases, $P^2IR$ in heterogeneous setting has the best performance. $P^2IR$ in homogeneous setting is better than deepwalk and node2vec in the same homogeneous setting, and is even better than *metapath2vec++* in heterogeneous setting (achieving the second best results). Overall, the superior performance of $P^2IR$ in Table 1 and 2 demonstrates the validity of our proposed universal graph embedding mechanism.

Table 2: F1-score (macro, micro) (%) of multi-label classification in heterogeneous graphs

| | training% | 10.00% | 20.00% | 30.00% | 40.00% | 50.00% | 60.00% | 70.00% | 80.00% | 90.00% |
|---|---|---|---|---|---|---|---|---|---|---|
| BlogCatalog (macro) | deepwalk | 45.13 ±0.68 | 44.64 ±0.21 | 44.52 ±0.42 | 44.64 ±0.23 | 44.32 ±0.21 | 44.36 ±0.46 | 44.78 ±0.48 | 44.33 ±0.62 | 44.49 ±0.86 |
| | node2vec | 45.78 ±0.68 | 45.42 ±0.30 | 45.28 ±0.32 | 45.41 ±0.18 | 45.17 ±0.20 | 45.19 ±0.36 | 45.57 ±0.13 | 45.04 ±0.31 | 44.96 ±0.55 |
| | metapath2vec++ | 37.46 ±0.61 | 36.72 ±0.36 | 36.69 ±0.29 | 36.58 ±0.28 | 36.74 ±0.17 | 36.90 ±0.33 | 36.89 ±0.32 | 36.42 ±0.41 | 36.16 ±0.98 |
| | **$P^2$IR** (Homogeneous) | 47.63 * ±3.16 | 50.99 * ±0.09 | 51.70 * ±0.19 | 50.04 * ±1.90 | 50.60 * ±1.73 | 50.34 * ±0.55 | 52.20 * ±1.11 | 51.88 * ±1.08 | 51.36 * ±0.26 |
| | **$P^2$IR** (Heterogeneous) | **49.65 ±0.63** | **51.47 ±0.40** | **52.69 ±0.24** | **53.37 ±0.45** | **53.73 ±0.01** | **53.97 ±0.43** | **53.83 ±0.62** | **54.07 ±0.40** | **53.36 ±0.84** |
| BlogCatalog (micro) | deepwalk | 47.93 ±0.48 | 47.36 ±0.15 | 47.25 ±0.45 | 47.30 ±0.19 | 47.07 ±0.16 | 47.09 ±0.48 | 47.39 ±0.49 | 47.02 ±0.72 | 47.27 ±0.82 |
| | node2vec | 48.52 ±0.50 | 48.20 ±0.18 | 48.01 ±0.31 | 48.18 ±0.17 | 47.97 ±0.25 | 48.04 ±0.39 | 48.22 ±0.15 | 47.83 ±0.32 | 47.95 ±0.55 |
| | metapath2vec++ | 40.90 ±0.34 | 40.06 ±0.21 | 40.10 ±0.25 | 39.97 ±0.22 | 40.04 ±0.17 | 40.22 ±0.29 | 40.21 ±0.22 | 39.82 ±0.44 | 39.66 ±0.82 |
| | **$P^2$IR** (Homogeneous) | 50.75 * ±2.74 | 54.26 * ±0.12 | 55.08 * ±0.08 | 53.41 * ±1.92 | 53.88 * ±0.50 | 53.57 * ±1.30 | 55.36 * ±1.07 | 54.93 * ±1.07 | 55.03 * ±0.52 |
| | **$P^2$IR** (Heterogeneous) | **53.06 ±0.45** | **54.77 ±0.21** | **55.93 ±0.17** | **56.41 ±0.31** | **56.86 ±0.06** | **57.28 ±0.38** | **57.13 ±0.43** | **57.43 ±0.14** | **56.98 ±0.53** |
| DBLP (macro) | Deepwalk | 74.48 * ±0.34 | 74.87 ±0.14 | 74.95 ±0.15 | 75.10 ±0.22 | 75.07 ±0.22 | 75.44 ±0.22 | 75.33 ±0.33 | 74.75 ±0.31 | 75.36 ±0.73 |
| | Node2vec | 73.37 ±0.24 | 73.94 ±0.11 | 74.00 ±0.18 | 74.25 ±0.23 | 74.06 ±0.31 | 74.52 ±0.21 | 74.52 ±0.35 | 74.32 ±0.26 | 74.55 ±0.57 |
| | Metapath2vec++ | **74.82 ±0.30** | 75.27 ±0.08 | 75.55 ±0.12 | 75.63 ±0.27 | 75.53 ±0.22 | 75.92 ±0.24 | 75.92 ±0.42 | 75.56 ±0.36 | 76.09 ±0.46 |
| | **$P^2$IR** (Homogeneous) | 72.51 ±0.91 | 76.89 * ±0.06 | 79.91 * ±0.09 | 82.14 * ±0.26 | 84.60 * ±0.42 | 86.34 * ±0.52 | 87.49 * ±0.31 | 88.21 * ±0.25 | 89.61 * ±0.58 |
| | **$P^2$IR** (Heterogeneous) | 74.06 ±0.37 | **78.43 ±0.61** | **81.00 ±0.19** | **83.18 ±0.16** | **84.95 ±0.22** | **86.91 ±0.54** | **88.30 ±0.29** | **89.18 ±0.17** | **90.37 ±0.38** |
| DBLP (micro) | deepwalk | 76.65 ±0.25 | 77.03 ±0.19 | 77.15 ±0.15 | 77.21 ±0.15 | 77.20 ±0.17 | 77.60 ±0.20 | 77.44 ±0.31 | 76.87 ±0.35 | 77.54 ±0.72 |
| | node2vec | 75.65 ±0.16 | 76.21 ±0.12 | 76.30 ±0.16 | 76.48 ±0.18 | 76.33 ±0.25 | 76.82 ±0.22 | 76.76 ±0.33 | 76.44 ±0.34 | 76.73 ±0.55 |
| | metapath2vec++ | 76.98 * ±0.21 | 77.38 ±0.10 | 77.66 ±0.08 | 77.70 ±0.21 | 77.61 ±0.18 | 78.04 ±0.18 | 77.95 ±0.39 | 77.54 ±0.36 | 78.02 ±0.47 |
| | **$P^2$IR** (Homogeneous) | 74.37 ±0.88 | 78.52 * ±0.09 | 81.47 * ±0.11 | 83.56 * ±0.28 | 85.81 * ±0.42 | 87.51 * ±0.54 | 88.44 * ±0.27 | 89.16 * ±0.22 | 90.54 * ±0.50 |
| | **$P^2$IR** (Heterogeneous) | **77.06 ±0.29** | **80.67 ±0.45** | **82.87 ±0.13** | **84.75 ±0.16** | **86.29 ±0.22** | **88.09 ±0.47** | **89.27 ±0.25** | **90.11 ±0.13** | **91.22 ±0.43** |

## 5 CONCLUSION AND FUTURE WORK

To summarize the whole work, we propose $P^2$IR, a general graph embedding solution with the novel notion of partial permutation invariant set function, in principle that it can capture an arbitrary dependence between neighbors and automatically decide the significance of neighbor nodes at different distance for both homogeneous and heterogeneous graphs. We provide a theoretical guarantee for the effectiveness of the whole model. Through conducting extensive experimental evaluation, we show $P^2$IR has better performance on both homogeneous and heterogeneous graphs, comparing to the stochastic trajectories based, matrix analytics based and graph neural network based state-of-the-art algorithms. For the future work, our model can be extended to more general cases, e.g., involving the rich content information out of graph neighborhood structures.

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

# Appendix

We provide the proof to Theorem 3.1 in the appendix. The proof in our paper borrows some idea from the proof sketch for a special case of our theorem in (Zaheer et al., 2017). While (Zaheer et al., 2017) only provides a small paragraph to explain the raw idea on how to prove a special case, we provide complete and rigorous proof and nontrivial extension to the more general case.

**Definition 5.1** (Power Sum Symmetric Polynomials). *For every integer $k \geq 0$, the $k$-th power sum symmetric polynomial in variables $x_1, x_2, \cdots, x_n$ is defined as*

$$p_k(x_1, x_2, \cdots, x_n) = \sum_{i=1}^{n} x_i^k.$$

**Definition 5.2** (Monomial Symmetric Polynomials). *For $\boldsymbol{\lambda} = [\lambda_1, \lambda_2, \ldots, \lambda_n]^\top \in \mathbb{R}^n$ where $\lambda_1 \geq \lambda_2 \geq \cdots \geq \lambda_n \geq 0$, let $\Omega_{\boldsymbol{\lambda}}$ be the set of all permutations of the entries in $\boldsymbol{\lambda}$. For $\boldsymbol{\lambda}$, the monomial symmetric polynomials is defined as*

$$m^{\boldsymbol{\lambda}} = \sum_{[\lambda_1, \cdots, \lambda_n]^\top \in \Omega_{\boldsymbol{\lambda}}} x_1^{\lambda_1} x_2^{\lambda_2} \cdots x_n^{\lambda_n}$$

**Lemma 5.3.** *If $f : \mathbb{X} \to \mathbb{R}$ is a continuous real-valued function defined on the compact set $\mathbb{X} \subset \mathbb{R}^n$, then $\forall \epsilon > 0$, there exists a polynomial function $p : \mathbb{X} \to \mathbb{R}$, such that $\sup_{x \in \mathbb{X}} |f(x) - p(x)| < \epsilon$.*

*Proof.* This lemma is a direct application of Stone-Weierstrass Theorem (Stone, 1937; 1948) for the compact Hausdorff space. $\square$

**Corollary 5.3.1.** *For any continuous real-valued function $f : \mathbb{X} \to \mathbb{R}$ of the form defined in Theorem 3.1,*

$$f(\underbrace{x_{1,1}, x_{1,2}, \cdots, x_{1,N_1}}_{G_1}, \underbrace{x_{2,1}, x_{2,2}, \cdots, x_{2,N_2}}_{G_2}, \cdots, \underbrace{x_{K,1}, x_{K,2}, \cdots, x_{K,N_K}}_{G_K}),$$

*there exist polynomial functions $p : \mathbb{X} \to \mathbb{R}$ also permutation invariant within each $G_k$ to closely approximate $f$.*

*Proof.* Let $\mathbb{S}^k$ be the set of all possible permutations of $\{1, \cdots, N_k\}$ for the group $G^k$, where $k \in \{1, 2, \ldots, K\}$. Suppose $\boldsymbol{\sigma}^k = [\sigma_1^k, \sigma_2^k, \ldots, \sigma_{N_k}^k] \in \mathbb{S}^k$ is some permutation for $G^k$. Then we know that $\forall \boldsymbol{\sigma}^k \in \mathbb{S}^k$ with $k \in \{1, 2, \ldots, K\}$, we have

$$f(\boldsymbol{x}_1, \boldsymbol{x}_2, \cdots, \boldsymbol{x}_K) = f([\boldsymbol{x}_1]_{\boldsymbol{\sigma}^1}, [\boldsymbol{x}_2]_{\boldsymbol{\sigma}^2}, \cdots, [\boldsymbol{x}_K]_{\boldsymbol{\sigma}^K})$$

where we let $\boldsymbol{x}_k$ to denote $[x_{k,1}, x_{k,2}, \cdots, x_{k,N_k}]$ for $G_k$ and $[\boldsymbol{x}_k]_{\boldsymbol{\sigma}^k}$ to denote the permuted $\boldsymbol{x}_k$ for simplicity. There are in total $\prod_{k=1}^{K}(N_k!)$ permutations.

By lemma 5.3, we can know that $\forall \epsilon > 0$, there exists a polynomial function $q : \mathbb{X} \to \mathbb{R}$ such that $\sup |f(\boldsymbol{x}_1, \boldsymbol{x}_2, \cdots, \boldsymbol{x}_K) - q(\boldsymbol{x}_1, \boldsymbol{x}_2, \cdots, \boldsymbol{x}_K)| \leq \epsilon$. This further implies that $\forall \boldsymbol{\sigma}^k \in \mathbb{S}^k$ with $k \in \{1, 2, \ldots, K\}$, we have

$$\sup |f([\boldsymbol{x}_1]_{\boldsymbol{\sigma}^1}, [\boldsymbol{x}_2]_{\boldsymbol{\sigma}^2}, \cdots, [\boldsymbol{x}_K]_{\boldsymbol{\sigma}^K}) - q([\boldsymbol{x}_1]_{\boldsymbol{\sigma}^1}, [\boldsymbol{x}_2]_{\boldsymbol{\sigma}^2}, \cdots, [\boldsymbol{x}_K]_{\boldsymbol{\sigma}^K})| \leq \epsilon$$

We let $p(\boldsymbol{x}_1, \boldsymbol{x}_2, \cdots, \boldsymbol{x}_K) = \frac{1}{\prod_{k=1}^K (N_k!)} \sum_{\boldsymbol{\sigma}^1 \in \mathbb{S}^1, \cdots, \boldsymbol{\sigma}^K \in \mathbb{S}^K} q([\boldsymbol{x}_1]_{\boldsymbol{\sigma}^1}, [\boldsymbol{x}_2]_{\boldsymbol{\sigma}^2}, \cdots, [\boldsymbol{x}_K]_{\boldsymbol{\sigma}^K})$ and by the property of permutation invariant within each group $G^k$ of the function $f$, we have that

$$\sup |f(\boldsymbol{x}_1, \boldsymbol{x}_2, \cdots, \boldsymbol{x}_K) - p(\boldsymbol{x}_1, \boldsymbol{x}_2, \cdots, \boldsymbol{x}_K)|$$

$$= \sup \left| f(\boldsymbol{x}_1, \boldsymbol{x}_2, \cdots, \boldsymbol{x}_K) - \frac{1}{\prod_{k=1}^K (N_k!)} \sum_{\boldsymbol{\sigma}^1 \in \mathbb{S}^1, \cdots, \boldsymbol{\sigma}^K \in \mathbb{S}^K} q([\boldsymbol{x}_1]_{\boldsymbol{\sigma}^1}, [\boldsymbol{x}_2]_{\boldsymbol{\sigma}^2}, \cdots, [\boldsymbol{x}_K]_{\boldsymbol{\sigma}^K}) \right|$$

$$= \sup \left| \frac{1}{\prod_{k=1}^K (N_k!)} \sum_{\boldsymbol{\sigma}^1 \in \mathbb{S}^1, \cdots, \boldsymbol{\sigma}^K \in \mathbb{S}^K} f([\boldsymbol{x}_1]_{\boldsymbol{\sigma}^1}, \cdots, [\boldsymbol{x}_K]_{\boldsymbol{\sigma}^K}) - \frac{1}{\prod_{k=1}^K (N_k!)} \sum_{\boldsymbol{\sigma}^1 \in \mathbb{S}^1, \cdots, \boldsymbol{\sigma}^K \in \mathbb{S}^K} q([\boldsymbol{x}_1]_{\boldsymbol{\sigma}^1}, \cdots, [\boldsymbol{x}_K]_{\boldsymbol{\sigma}^K}) \right|$$

$$\leq \frac{1}{\prod_{k=1}^K (N_k!)} \sum_{\boldsymbol{\sigma}^1 \in \mathbb{S}^1, \cdots, \boldsymbol{\sigma}^K \in \mathbb{S}^K} \sup \left| f([\boldsymbol{x}_1]_{\boldsymbol{\sigma}^1}, [\boldsymbol{x}_2]_{\boldsymbol{\sigma}^2}, \cdots, [\boldsymbol{x}_K]_{\boldsymbol{\sigma}^K}) - q([\boldsymbol{x}_1]_{\boldsymbol{\sigma}^1}, [\boldsymbol{x}_2]_{\boldsymbol{\sigma}^2}, \cdots, [\boldsymbol{x}_K]_{\boldsymbol{\sigma}^K}) \right|$$

$$\leq \frac{1}{\prod_{k=1}^K (N_k!)} \sum_{\boldsymbol{\sigma}^1 \in \mathbb{S}^1, \cdots, \boldsymbol{\sigma}^K \in \mathbb{S}^K} \epsilon$$

$$= \epsilon$$

Therefore, we can see that the polynomial function $p(\boldsymbol{x}_1, \boldsymbol{x}_2, \cdots, \boldsymbol{x}_K)$ is permutation invariant within each group and can closely approximate the function $f(\boldsymbol{x}_1, \boldsymbol{x}_2, \cdots, \boldsymbol{x}_K)$.

$\square$

**Lemma 5.4.** *Suppose $p : \mathbb{X} \to \mathbb{R}$ is a real-valued polynomial in the form*

$$p(\underbrace{x_{1,1}, x_{1,2}, \cdots, x_{1,N_1}}_{G_1}, \cdots, \underbrace{x_{K,1}, x_{K,2}, \cdots, x_{K,N_K}}_{G_K}),$$

*which is permutation invariant within each group $G^k$, $k \in \{1, 2, \ldots, K\}$. Then, $p$ can be represented as*

$$p(\underbrace{x_{1,1}, x_{1,2}, \cdots, x_{1,N_1}}_{G_1}, \cdots, \underbrace{x_{K,1}, x_{K,2}, \cdots, x_{K,N_K}}_{G_K}) = \sum_i c_i m_1^{\boldsymbol{\lambda}_1^i} m_2^{\boldsymbol{\lambda}_2^i} \ldots m_K^{\boldsymbol{\lambda}_K^i}$$

*where $c_i$ is the rational coefficient, $m_k^{\boldsymbol{\lambda}_K^i}$ denotes a monomial symmetric polynomial of variables $x_{k,1}, x_{k,2}, \cdots, x_{k,N_k}$ within group $G_k$ and $\boldsymbol{\lambda}_K^i \in \mathbb{R}^n$ are some certain exponents for the monomial symmetric polynomial.*

*Proof.* Suppose that the polynomial is expressed as a summation of monomials

$$p(\underbrace{x_{1,1}, x_{1,2}, \cdots, x_{1,N_1}}_{G_1}, \cdots, \underbrace{x_{K,1}, x_{K,2}, \cdots, x_{K,N_K}}_{G_K}) = \sum_r c_r \prod_{j=1}^{N_1} x_{1,j}^{\alpha_{1,j}^r} \prod_{j=1}^{N_2} x_{2,j}^{\alpha_{2,j}^r}, \cdots, \prod_{j=1}^{N_K} x_{K,j}^{\alpha_{K,j}^r}$$

where $\alpha_{k,j}^r$ is the exponent of $x_{k,j}$ for the $r$-th monomial.

We consider the term $c_r \prod_{j=1}^{N_1} x_{1,j}^{\alpha_{1,j}^r} \prod_{j=1}^{N_2} x_{2,j}^{\alpha_{2,j}^r} \cdots \prod_{j=1}^{N_K} x_{K,j}^{\alpha_{K,j}^r}$ for a certain $r$ in the above summation. Since $p(\cdot)$ is partially symmetric within the group $G_1$, then there must exist terms having exponents of permuted $\alpha_{1,j}^r, \forall j \in \{1, \ldots, N_1\}$ on $x_{1,j}$ while the other factors $c_r \prod_{j=1}^{N_2} x_{2,j}^{\alpha_{2,j}^r} \cdots \prod_{j=1}^{N_K} x_{K,j}^{\alpha_{K,j}^r}$ remain the same. (Otherwise, the polynomial is not permutation-invariant w.r.t. group $G_1$.) By summing up those terms together, we get a term with a factor of monomial symmetric polynomial as $c_r m_1^{\boldsymbol{\alpha}_1^r} \prod_{j=1}^{N_2} x_{2,j}^{\alpha_{2,j}^r} \cdots \prod_{j=1}^{N_K} x_{K,j}^{\alpha_{K,j}^r}$. Based on this term, we consider that the polynomial $p(\cdot)$ is also permuted invariant within $G_2$, which implies that there must exist terms having exponents of permuted $\alpha_{2,j}^r, \forall j \in \{1, \ldots, N_2\}$ on $x_{2,j}$ while the other factors $c_r m_1^{\boldsymbol{\alpha}_1^r} \prod_{j=1}^{N_3} x_{3,j}^{\alpha_{3,j}^r} \cdots \prod_{j=1}^{N_K} x_{K,j}^{\alpha_{K,j}^r}$ remain the same. Therefore, adding up all those terms together, we have $c_r m_1^{\boldsymbol{\alpha}_1^r} m_2^{\boldsymbol{\alpha}_2^r} \prod_{j=1}^{N_3} x_{3,j}^{\alpha_{3,j}^r} \cdots \prod_{j=1}^{N_K} x_{K,j}^{\alpha_{K,j}^r}$. Carrying out the above procedures recursively, we can eventually have $c_r m_1^{\boldsymbol{\alpha}_1^r} m_2^{\boldsymbol{\alpha}_2^r} \ldots m_K^{\boldsymbol{\alpha}_K^r}$. For all the remaining terms in the polynomial $p(\cdot)$, performing the same steps will lead to completion of our proof. $\square$

**Example 5.5.** $p(x_1^1, x_2^1, x_1^2, x_2^2)$ *is a polynomial that is permutation invariant within each group,* $G_1 = \{x_1^1, x_2^1\}$ *and* $G_2 = \{x_1^2, x_2^2\}$. *We let*

$$
\begin{aligned}
p(x_{1,1}, x_{1,2}, x_{2,1}, x_{2,2}) =& x_{1,1}x_{1,2}^2 x_{2,1}x_{2,2}^2 + x_{1,1}^2 x_{1,2}x_{2,1}^2 x_{2,2} + x_{1,1}x_{1,2}^2 x_{2,1}^2 x_{2,2} + x_{1,1}^2 x_{1,2}x_{2,1}x_{2,2}^2 \\
& + x_{1,1}^2 x_{1,2}^3 x_{2,1}^3 x_{2,2}^4 + x_{1,1}^3 x_{1,2}^2 x_{2,1}^4 x_{2,2}^3 + x_{1,1}^2 x_{1,2}^3 x_{2,1}^4 x_{2,2}^3 + x_{1,1}^3 x_{1,2}^2 x_{2,1}^3 x_{2,2}^4.
\end{aligned}
$$

*It is easy to observe that* $p(x_{1,1}, x_{1,2}, x_{2,1}, x_{2,2})$ *can be rewritten as*

$$
\begin{aligned}
p(x_{1,1}, x_{1,2}, x_{2,1}, x_{2,2}) =& (x_{1,1}x_{1,2}^2 + x_{1,1}^2 x_{1,2})(x_{2,1}x_{2,2}^2 + x_{2,1}^2 x_{2,2}) \\
& + (x_{1,1}^2 x_{1,2}^3 + x_{1,1}^3 x_{1,2}^2)(x_{2,1}^3 x_{2,2}^4 + x_{2,1}^4 x_{2,2}^3) \\
=& m_1^{(2,1)} m_2^{(2,1)} + m_1^{(3,2)} m_2^{(4,3)}.
\end{aligned}
$$

**Lemma 5.6.** *For a symmetric polynomial* $p(x_1, \ldots, x_n)$ *of n variables, it can be expressed by a polynomial in the power sum symmetric polynomials* $p_k(x_1, \ldots, x_n)$ *for* $1 \le k \le n$ *with rational coefficients.*

*Proof.* This lemma is a direct result of the fact that $p_1, p_2, \ldots, p_n$ are algebraically independent and the ring of symmetric polynomials with rational coefficients can be generated as a $\mathbb{Q}$-algebra, i.e. $\mathbb{Q}[p_1, p_2, \ldots, p_n]$. (Macdonald, 1998; Stanley, 2001) This lemma can also be easily proved by the combination of the fundamental theorem of symmetric polynomials and newton's identities. $\square$

**Proof of Theorem 3.1**

*Proof.* By Corollary 5.3.1, for any function $f : \mathbb{X} \to \mathbb{R}$ that are permutation-invariant w.r.t. the variables within each group $G_k, \forall k \in \{1, 2, \ldots, K\}$, which is in the form

$$
f(\underbrace{x_{1,1}, x_{1,2}, \cdots, x_{1,N_1}}_{G_1}, \underbrace{x_{2,1}, x_{2,2}, \cdots, x_{2,N_2}}_{G_2}, \cdots, \underbrace{x_{K,1}, x_{K,2}, \cdots, x_{K,N_K}}_{G_K}),
$$

we can always find a polynomial function $p : \mathbb{X} \to \mathbb{R}$ that are also permutation-invariant w.r.t. the variables within each group $G_k$ to approximate $f$ closely in a given small error tolerance $\epsilon$.

Here we first define the function $g_k : \mathbb{R} \to \mathbb{R}^{N_k}$ in the following form,

$$
g_k(x_{k,n}) = \begin{bmatrix} x_{k,n} \\ x_{k,n}^2 \\ x_{k,n}^3 \\ \vdots \\ x_{k,n}^{N_k} \end{bmatrix}
$$

which thus leads to

$$
\sum_{n=1}^{N_k} g_k(x_{k,n}) = \begin{bmatrix} \sum_{n=1}^{N_k} x_{k,n} \\ \sum_{n=1}^{N_k} x_{k,n}^2 \\ \sum_{n=1}^{N_k} x_{k,n}^3 \\ \vdots \\ \sum_{n=1}^{N_k} x_{k,n}^{N_k} \end{bmatrix} = \begin{bmatrix} p_1(x_{k,1}, \cdots, x_{k,N_k}) \\ p_2(x_{k,1}, \cdots, x_{k,N_k}) \\ p_3(x_{k,1}, \cdots, x_{k,N_k}) \\ \vdots \\ p_{N_k}(x_{k,1}, \cdots, x_{k,N_k}) \end{bmatrix}
$$

Therefore, we generate a sequence of power sums basis by $\sum_{n=1}^{N_k} g_k(x_{k,n})$.

By Lemma 5.4, the polynomial function $p(x_{1,1}, x_{1,2}, \cdots, x_{1,N_1}, \cdots, x_{K,1}, x_{K,2}, \cdots, x_{K,N_K})$ can be expressed as $\sum_{i=1} c_i m_1^{\boldsymbol{\lambda}_1^i} m_2^{\boldsymbol{\lambda}_2^i} \ldots m_K^{\boldsymbol{\lambda}_K^i}$. Note that each $m_k^{\boldsymbol{\lambda}_k^i}$ is a symmetric polynomial, which thus can be rewritten as a polynomial expression of power sum basis,

$$
p_1(x_{k,1}, \cdots, x_{k,N_k}), p_2(x_{k,1}, \cdots, x_{k,N_k}), \cdots, p_{N_k}(x_{k,1}, \cdots, x_{k,N_k}).
$$

which has been generated by $\sum_{n=1}^{N_k} g_k(x_{k,n})$.

Hence the polynomial $p(x_{1,1}, x_{1,2}, \cdots, x_{1,N_1}, \cdots, x_{K,1}, x_{K,2}, \cdots, x_{K,N_K})$ is also a function of $\sum_{n=1}^{N_k} g_k(x_{k,n}), \forall k \in \{1, 2, \ldots, K\}$, which will be expressed as

$$p(x_{1,1}, x_{1,2}, \cdots, x_{1,N_1}, \cdots, x_{K,1}, x_{K,2}, \cdots, x_{K,N_K}) = h(\sum_{n=1}^{N_1} g_1(x_{1,n}), \sum_{n=1}^{N_2} g_2(x_{2,n}), \cdots, \sum_{n=1}^{N_K} g_K(x_{K,n})).$$

Thus, the function $f$ could be approximate in any given error tolerance $\epsilon$ by a polynomial $h(\sum_{n=1}^{N_1} g_1(x_{1,n}), \sum_{n=1}^{N_2} g_2(x_{2,n}), \cdots, \sum_{n=1}^{N_K} g_K(x_{K,n}))$, which finishes our proof.

$\square$

