# OpenReview forum: "P^2IR: Universal Deep Node Representation via Partial Permutation Invariant Set Functions"
_ICLR.cc/2019/Conference_

### Official Review · AnonReviewer3 · 2018-10-19
**The paper is good but quality of experiments can be further improved.**

**Rating:** 5
**Confidence:** 5

**Review:**

The authors introduce the idea of a partial permutation invariant set function and use this to learn node embeddings. The paper is well-written and the discussion is quite easy to follow along. The paper also introduces some interesting concepts like a partial permutation invariant set function. However, I find that some of the paper's main claims can be further backed up by more experiments.

The paper is well-written. The authors evaluate their approach on a large number of real-world homogeneous and heterogeneous graphs. Furthermore, they show the stability of their method by showing consistently good results even when training set size is varied. Defined the notion of a partial permutation invariant set function and provided theoretical guarantees pertaining to this.

One of the strengths of the proposed method is its ability to "automatically decide the significance of nodes at different distances." The authors devote a good portion of their paper to talk about this. However, a recent paper published in NIPS '18 [1] with a pre-print available much earlier solves this problem by applying attention over powers of a transition matrix. The authors should talk about [1] and ideally compare against them.

I feel that a lot of the authors main claims can be strengthened further if more experiments were shown to back these up. What I mean to say is, it would be nice to see some other experiments apart from just classification performance. To compare with [1], for instance, they show link prediction/classification results but on top of these they also show that their method may choose very different "neighborhoods" to attend to.

The authors gave a fairly comprehensive review of related literature (which was good!) and they mentioned that "most existing methods either explicitly or implicitly restrict the dependence form of each node to its neighbors and also the depth of neighbors." I feel another approach they should compare against which does not seem to have this problem is [2] since the method learns "role-based" embeddings which are more dependent on structure rather than proximity.

The paper in its current form is fairly good. If comparison can be made against [1] & [2] and some additional experiments can be added, the quality of the paper can be improved further.
[1] Watch Your Step: Learning Node Embeddings via Graph Attention. Abu-El-Haija et al. In Proc. of NIPS 2018.
[2] Higher-Order Network Representation Learning. Rossi et al. In Proc. of WWW 2018.

There are some minor errors in the paper:

Table 1 in page 7 seems to be an error. It's an empty table and it is not referred to anywhere in the paper.

The format of some references needs double-checking. For example,

"Jian Tang, Meng Qu, and Qiaozhu Mei. Pte: Predictive text embedding through large-scale heterogeneous text networks. In Proceedings of the 21th ACM SIGKDD International Conference on Knowledge Discovery and Data Mining, pp. 1165–1174. ACM, 2015a."

(1) "21th" should be "21st".

"Shiyu Chang, Wei Han, Jiliang Tang, Guo-Jun Qi, Charu C Aggarwal, and Thomas S Huang. Heterogeneous network embedding via deep architectures. In Proceedings of the 21th ACM SIGKDD International Conference on Knowledge Discovery and Data Mining, pp. 119–128. ACM, 2015."

(2) "21th" should be "21st".

"Mingdong Ou, Peng Cui, Jian Pei, Ziwei Zhang, and Wenwu Zhu. Asymmetric transitivity preserving graph embedding. In Proceedings of the 22Nd ACM SIGKDD International Conference on Knowledge Discovery and Data Mining, KDD ’16, pp. 1105–1114, 2016."

(3) "22Nd" should be "22nd"

"Daixin Wang, Peng Cui, and Wenwu Zhu. Structural deep network embedding. In Proceedings of the 22Nd ACM SIGKDD International Conference on Knowledge Discovery and Data Mining, KDD’16, pp. 1225–1234, 2016."

(4) "22Nd" should be "22nd"

"Bryan Perozzi, Rami Al-Rfou, and Steven Skiena. Deepwalk: Online learning of social representations. In Proceedings of the 20th ACM SIGKDD international conference on Knowledge discovery and data mining, pp. 701–710. ACM, 2014"

(5) "international conference on Knowledge discovery and data mining" should be "International Conference on Knowledge Discovery and Data Mining"

"Tomas Mikolov, Ilya Sutskever, Kai Chen, Greg S Corrado, and Jeff Dean. Distributed representations of words and phrases and their compositionality. In Advances in neural information processing systems, pp. 3111–3119, 2013."

(6) "Advances in neural information processing systems" should be "Advances in Neural Information Processing Systems"

"Jian Tang, Meng Qu, Mingzhe Wang, Ming Zhang, Jun Yan, and Qiaozhu Mei. Line: Large-scale information network embedding. In Proceedings of the 24th International Conference on World Wide Web, pp. 1067–1077. International World Wide Web Conferences Steering Committee, 2015b."

(7) "International World Wide Web Conferences Steering Committee" should be removed.

The 3rd sentence in the abstract is a little bit too long. It would be better if the authors could break the sentence into shorter ones. Here is the sentence: "While most existing approaches rely on defining the specific neighborhood dependence as the computation mechanism of representations, which may exclude important subtle structures within the graph and dependence among neighbors, we propose a novel graph node embedding method (namely P2IR) via developing a novel notion, namely partial permutation invariant set function."

---

### Official Review · AnonReviewer1 · 2018-11-03
**Interesting idea, missing related work, missing results discussion and overall poor presentation**

**Rating:** 5
**Confidence:** 3

**Review:**

The paper explores the very interesting and relevant problem of universal node representation.  It points out that although powerful models for representation learning on graphs exists, most existing works require to pre-define a pairwise node similarity or to specify model parameters. Hence, the authors propose a novel model that doesn’t require to pre-define neighbors nor to specify the dependence form between each node and its neighbors.


Pros:
- This work studies the important question of universal node embedding model that require minimal user-defined specifications.
- It proposes an original and novel solution to achieve universal node embedding based on partially permutation invariant function.
- Provides theoretical guarantee.

Cons:
- Some recent works on structural node embedding are directly related to this work but missing in the related work section: struc2vec [1] and GraphWave [2].
- In the experiment section, it would be necessary to provide the values of the tuned hyper-parameters for each model for reproducibility.
- The results are not really analysed nor discussed beyond noticing that P^2IR performs better than other models in most cases. For instance, the authors don't discuss the complexity of the different models, or don’t give intuition as to whether the improvements are significant.
- It would be relevant to include some node embeddings models (such as [1,2]) in the baseline methods as they have been shown to outperform node2vec/deepwalk in some classification tasks.

minor comments on the text:
- on page 2, WFS instead of BFS
- on page 5, please spell out 'NN function'
- on page 6, in the last equation characterizing the mapping of node v, it is not clear why the subscript k  in phi_k is there. (similarly for eq. (3) and (4) and subsequent mention of phi).
- on page 7, Table 1 is useless.

1. Ribeiro, L. F., Saverese, P. H., and Figueiredo, D. R. (2017). Struc2vec: Learning node representations from structural identity
2. Donnat, C., Zitnik, M., Hallac, D., and Leskovec, J. (2018). Learning structural node embeddings via diffusion wavelets.

---

### Official Review · AnonReviewer2 · 2018-11-04
**A unified way to incorporate high-order proximity information for graph embedding**

**Rating:** 7
**Confidence:** 4

**Review:**

The paper proposes a formulation for taking care of neighborhood of different distances for graph embedding. It makes use of a notion called permutation invariant function which defined as a function where if we swap any features in the inputs, the function value remains the same. Given this, they make two contributions to make the consideration of neighborhood of different distances for graph embedding possible. First, they make the assumption that the contribution of neighbours of same steps should be the same and thus permutable in defining how the embedding function of a node is depending on this neighbours. Another one is the use of 1-d NN for estimating the contribution from 1-step, 2-step and up to infinite-step. Then, the overall problem formulation is defined and can be learned using SDG.

+ve:
1. The paper is well organized and clearly presented.
2. The technique proposed can handle neighborhood of different distances while the existing methods make explicit or implicit assumptions (and thus restrictions) about the neighborhood to be considered.
3. The proposed method performs consistently better than a number of representative deep graph models based on a number of benchmark datasets.
4. The method is applicable to both homogeneous and heterogeneous graphs.

-ve:
1. The part after Eq.(4) and before Section 3.3 is important but a bit hard to read as compared to the other parts of the paper.
2. The graphs tested are not particular large. Larger ones should be tested.
3. The methods being compared are not the most recent ones (all published in 2016 or before).
4. Something wrong with Table 1?

---

### Official Review · AnonReviewer4 · 2018-11-07
**Computational costly heterogeneous graph embedding**

**Rating:** 4
**Confidence:** 4

**Review:**

This paper proposed a heterogeneous graph embedding method P^2IR. The author(s) first
argued that such an embedding should be invariant to partial permutations of nodes.
Then the authors gave a general formulation of such an embedding in theorem 3.1.
Then the authors instantiated this general formulation by a neural network
parametrization, which can be optimized based on the L^2 loss and a supervised
regularizer. The method is tested against graph embedding methods that do not
need node attributes (in GCN the authors "eliminated" the node attributes) on
semi-supervised node classification tasks, showing a significant improvement.

My main criticism is that the authors did not clarify or put any efforts on
solve the high computational complexity.
The proposed method needs to perform a spectral decomposition of
the adjacency matrix, which has cubic complexity. This is unacceptable,
making the method less useful for real networks.
Furthermore, to optimize the embedding using SGD requires graph
Fourier transformations that have quadratic complexity.
In section 3.3, the exact complexity should be given, without which
the technique is incomplete.

An important reference "Graph Attention Networks. P. Veličković et al. 2018."
is missing, which has the similar idea to automatically learn the neighborhood
proximities. It should be cited as this is a key idea to motivate the paper.

The presentation quality is not satisfactory. For example, in page 3, f() has K
matrix arguments, then in page 4 theorem 3.1, f() takes KN arguments.
Please make it consistent.

In page 5, the formulations from eq.(3) to eq.(5) can be further unified
and simplified. From eq.(3) to eq.(4) is not straightforward and need more
explanations. If you use \mathcal{R} as the embedding, it should appear in
eq.(4) to be consistent.

Table 1 has no contents.

In the experimental results, the performance of GCN with node attributes
should be given for completeness (although the comparison is less fair).
A related question is how to incorporate node attributes in your framework?

In the heterogeneous experiments, for completeness, the authors are suggested
to compare against a heterogeneous version of GCN (again, with and without node
attributes) such as "Modeling Relational Data with Graph Convolutional Networks.
Schlichtkrull et al. 2017."

The paper is longer than the recommended length.

---

### Meta-Review · Area_Chair1 · 2018-12-14
**Evaluations, complexity, comparisons to the most recent methods.**

**Confidence:** 5
**Recommendation:** Reject

**Metareview:**

AR1 is concerned with the presentation of the paper and the complexity as well as missing discussion on recent  embedding methods. AR2 is concerned about comparison to recent methods and the small size of datasets.  AR3 is also concerned about limited comparisons and evaluations. Lastly, AR4 again points out the poor complexity due to the spectral decomposition. While authors argue that the sparsity can be exploited to speed up computations, AR4 still asks for results of the exact model with/without any approximation, effect of clipping spectrum, time complexity versus GCN, and more empirical results covering all these aspects. On balance, all reviewers seem to voice similar concerns which need to be resolved. However, this requires more than just a minor revision of the manuscript. Thus, at this time, the proposed paper cannot be accepted.